# Label-Free Saliva Test for Rapid Detection of Coronavirus Using Nanosensor-Enabled SERS

**DOI:** 10.3390/bioengineering10030391

**Published:** 2023-03-22

**Authors:** Swarna Ganesh, Ashok Kumar Dhinakaran, Priyatha Premnath, Krishnan Venkatakrishnan, Bo Tan

**Affiliations:** 1Keenan Research Center for Biomedical Science, Unity Health Toronto, Toronto, ON M5B 1W8, Canada; 2Institute for Biomedical Engineering, Science and Technology (I BEST), Partnership between Toronto Metropolitan University and St. Michael’s Hospital, Toronto, ON M5B 1W8, Canada; 3Ultrashort Laser Nanomanufacturing Research Facility, Department of Mechanical and Industrial Engineering, Toronto Metropolitan University, 350 Victoria Street, Toronto, ON M5B 2K3, Canada; 4Department of biomedical engineering, College of Engineering and Applied Sciences, University of Wisconsin, Milwaukee, WI 53211, USA; 5Nanocharacterization Laboratory, Department of Aerospace Engineering, Toronto Metropolitan University, 350 Victoria Street, Toronto, ON M5B 2K3, Canada

**Keywords:** COVID-19, nanosensors, SERS, coronavirus

## Abstract

The recent COVID-19 pandemic has highlighted the inadequacies of existing diagnostic techniques and the need for rapid and accurate diagnostic systems. Although molecular tests such as RT-PCR are the gold standard, they cannot be employed as point-of-care testing systems. Hence, a rapid, noninvasive diagnostic technique such as Surface-enhanced Raman scattering (SERS) is a promising analytical technique for rapid molecular or viral diagnosis. Here, we have designed a SERS- based test to rapidly diagnose SARS-CoV-2 from saliva. Physical methods synthesized the nanostructured sensor. It significantly increased the detection specificity and sensitivity by ~ten copies/mL of viral RNA (~femtomolar concentration of nucleic acids). Our technique combines the multiplexing capability of SERS with the sensitivity of novel nanostructures to detect whole virus particles and infection-associated antibodies. We have demonstrated the feasibility of the test with saliva samples from individuals who tested positive for SARS-CoV-2 with a specificity of 95%. The SERS—based test provides a promising breakthrough in detecting potential mutations that may come up with time while also preparing the world to deal with other pandemics in the future with rapid response and very accurate results.

## 1. Introduction

Viruses evolve swiftly and unpredictably, challenging the effectiveness of disease diagnostics. The extensive availability of accurate and rapid testing procedures is precious in unraveling the complex dynamics involved in SARS-CoV-2 infection and immunity. A substantial fraction of viral transmission occurs before the individuals develop symptoms [1] or from an asymptomatic individual. Nucleic Acid Testing (NATs) is predominantly performed only on symptomatic individuals, primarily due to the lack of data on kinetics and sensitivity testing on asymptomatic individuals [1]. Hence, large-scale diagnostic testing is a crucial tool for containing outbreaks such as COVID-19 in the future. 

Currently, available detection technologies are primarily based on molecular techniques such as RT-PCR and its spinoff technology [2] and serological techniques such as fluorescent antibody detection [3], ELISA, and lateral flow assays [4]. However, the application of the existing detection technologies is limited due to the dependence on complex operations, high-cost requirements, specialized equipment types, and low sensitivity [5]. Furthermore, RT-PCR tests for SARS-CoV-2 have a detection sensitivity of 500–1000 viral copies/mL [6], which reduces the applicability for presymptomatic and asymptomatic individuals, where is viral load is very low. The low viral load could also lead to false negatives since the viral load is beyond the analytical sensitivity of the existing diagnostic techniques [7]. The serological diagnostic tests also have not shown to be highly effective due to the high false positives and cross-reactivity with other seasonal, structurally, and symptomatically similar viruses [8]. Moreover, PCR and serological testing for evolving infectious diseases from novel pathogens or mutations in existing pathogens are exclusively based on pathogen-specific molecular probes such as oligomers and antibodies. So, developing a rapid, simple, label-free, cost-effective virus diagnostic technology would aid early clinical intervention and reduce transmission. 

Surface-enhanced Raman scattering (SERS), a versatile, noninvasive, and rapid detection technology, has the potential to be developed as a rapid diagnostic test for SARS-CoV-2. SERS- based approach has multiple advantages over conventional diagnostic techniques, including high sensitivity and minimal sample preparation, which could be adapted to test a wide range of biological molecules and pathogens in their native state in saliva, serum, and blood samples [9,10]. 

Recent studies applying vibrational spectroscopy for SARS-CoV-2 detection from saliva have been reported [11]. However, although the use of Raman spectroscopy provides the unique vibrational fingerprint of the pathogen, the signal enhancement in the absence of a nanosensor is marginal. Furthermore, it is severely affected by the inherent noise associated with Raman spectroscopic measurements [12,13]. Hence, to apply versatile technology such as Raman spectroscopy, using nanosensors is essential to enhance the unique signals associated with pathogens without ambiguity. 

Utilizing SERS for pathogen detection requires a nanosensor with high sensitivity, reproducibility, and high signal enhancement [14]. Conventionally used nanosensors utilizing noble metals such as gold, silver, and other coinage metals rely on SERS hot spots for signal enhancements [15]. However, despite the single-molecule sensitivity and high signal enhancement, the conventional SERS substrates often yield poor detection reliability for pathogens due to a lack of reproducible signals [16]. To achieve reliable signal enhancement, researchers adopted labels specific to SARS-CoV-2, such as spike protein peptides [17,18,19], which capture the SARS-CoV-2 viral particles. However, a rapidly mutating virus such as SARS-CoV-2 requires a label-free detection approach capable of detecting a broad range of mutant strains. Hence, a rapid, label-free diagnostic platform that can be manufactured for large–scale deployment is essential to managing future pandemics. 

Here, we report a rapid, label-free saliva test for SARS-CoV-2 using Surface-enhanced Raman scattering technology. The nanostructured sensor was synthesized using the physical process of laser-assisted multiphoton ionization of Silicon. By adopting the physical synthesis process, we have eliminated the use of noble metals, which yielded high reproducibility and repeatability, as demonstrated by a low signal-to-noise ratio and <10% relative standard deviation of the amplified signals. By utilizing the ultra-high sensitivity of the nanostructured sensor, we established the SERS fingerprint of the structural components of the SARS-CoV-2 virus with a limit of detection as low as a single molecule of viral RNA, spike protein, and nucleocapsid protein. Furthermore, the high sensitivity of the nanostructured sensor, combined with a simple machine-learning algorithm, enabled the detection of SARS-CoV-2 directly from clinical saliva samples with an accuracy of 95.6%. We validated the specificity of the saliva test through cross-reactivity analysis with three different viral strains (Influenza, HCOV OC43, RSV). We achieved a specificity of 100% for SARS-CoV-2 detection. The results obtained from SERS based saliva test were validated with a conventional RT- PCR assay. We envision that our high-performance COVID-19 rapid test using saliva will facilitate the rapid deployment of low-cost and efficient COVID-19 testing in an extensive population screening. 

## 2. Materials and Methods

### 2.1. Synthesis of Nanostructured Sensor

The nanostructured sensor was synthesized using a Clark-MXR IMPULSE Yb-doped fiber amplified ultrashort pulsed laser. The pulsed laser system generated a laser plume to initiate multiphoton ionization of the silicon wafer. The Silicon wafer used to generate the nanostructured sensor was a 0.02 Ω cm p-type silicon (100) wafer obtained from Universal Wafers. The laser wavelength was maintained at 1030 nm, dwell time (0.5 ms), laser power (14 W), and pulse width (214 fs) were maintained constant. In order to maintain the consistency of the nanostructure formation, the nanostructures were generated using a 300 × 300-line array; a piezo-driven raster stage was used to move the laser beam across the substrate using an array designed using the EzCAD software(V 2.7.6). 

The nanostructured sensor was made using a physical synthesis process of multiphoton ionization. The contact of an ultrashort pulsed laser with a solid single crystalline silicon wafer entails highly complex physical and chemical rearrangement processes resulting in the formation of the three-dimensional arranged nanosensor with unique chemical composition. During the interaction of a femtosecond laser pulse with the surface of solid single-crystalline Silicon, the temperature of the surrounding environment increases rapidly, resulting in the vaporization of the Si ions and the formation of a high-temperature plasma plume [20]. The ions in the plasma plume primarily comprise Si ions (evaporated from the Si) and Oxygen ions from the surrounding atmospheric air. The interaction of ionic species in the plume leads to nanoparticle cluster formation, which condenses on the surface of the silicon substrate [21]. These nanoparticle clusters are arranged three-dimensionally on the surface of Silicon. The physical synthesis process of multiphoton ionization further incorporates the oxygen ions from the surrounding plume environment as functional groups on the surface of the nanostructured sensor, as reported in previous works [22,23].

### 2.2. SERS Spectral Analysis

To acquire the SERS spectra of SARS-CoV-2 RNA (ATCC product code: VR-1986D), SARS-CoV-2 spike protein (Bio Vision Inc., Milpitas, CA, USA), Nucleocapsid protein (Life Sensors Inc., Malvern, PA, USA), 10 µL of the analyte was placed onto the nanostructured sensor. The SERS spectra were obtained using a portable Raman spectrometer from the B&W Tek NanoRam system. The Raman excitation laser of 785 nm with 350 mW power was used to obtain the SERS data. The acquisition time was kept constant for 30 s, and the spectra were obtained in triplicates. The SERS peaks used for analysis were carefully chosen with a signal-to-noise ratio higher than 3 times the standard deviation of the data. Further, to ensure the reproducibility and uniformity of signals, the acquisition time was kept constant at 10 s, and 3 acquisitions were made.

### 2.3. Cell Culture

The following viruses and the corresponding host cell line were obtained from ATCC: Human Coronavirus (OC43), Respiratory syncytial virus (RSV), and Influenza. The viruses and the host cells were cultured according to the protocol specified by ATCC. The viruses were harvested according to the manufacturer’s protocol. The viral load was calculated using plaque assay [24]. 

### 2.4. Clinical Saliva Samples

The saliva samples were obtained from Biocollections Worldwide Inc. The study has been approved by Ryerson Ethics Board (REB-2020-350-1). The samples were collected using traditional methods for developing molecular assays. Informed consent was obtained from the participants. The total number of subjects involved in the study was 30 (SARS-CoV-2 positive: 20, SARS-CoV-2 negative: 10). The samples were prepared by mixing 5 µL of saliva with a 5 µL of buffer volume to obtain the SERS spectra. 

### 2.5. Data Analysis

All the statistical analyses were performed using GraphPad Prism (V 8.0.2). We used a two-tailed *t*-test and one-way ANOVA to compare the differences between the sample groups; *p* < 0.05 was considered statistically significant. 

### 2.6. Machine Learning Algorithm

The preprocessed SERS spectra were dimensionally reduced to principal components using PCA. The PCs with maximum variances were used to train the decision tree algorithm. The decision tree algorithm splits the data into nodes based on class purity. The decision tree parameters used for the classification were: the minimum number of instances in the leaves was maintained at 3, and the maximal tree depth was maintained at 5 to limit non-specific sub-set classification. Further, the algorithm was instructed to stop classification after the majority threshold of 95% was reached. The machine learning algorithm’s performance was assessed using the calibrated probabilities of classification. The classification threshold was maintained at 0.5.

## 3. Results

### Saliva–Based Label-Free SERS Assay for Rapid SARS-CoV-2 Diagnosis

Nasal swab molecular assays have remained the gold standard of testing for SARS-CoV-2 detections. Still, they require substantial technical expertise and expensive equipment and are time-consuming. Here, the label-free saliva-based SARS-CoV-2 test uses a streamlined process that does not require specialized equipment beyond a portable Raman spectrometer (Figure 1). The test is performed by collecting the saliva sample to place 10 µL of the sample mixed with buffer onto the nanostructured sensor for label-free SERS detection. The nanostructured sensor was fabricated using the physical synthesis process of multiphoton ionization using an ultrashort pulsed laser [21]. The physical characterization of nanostructured sensors showed that the sensors are <10 nm, which has substantially improved the sensor’s SERS sensitivity [22]. Additionally, the surface functional groups on the Surface of the nanostructured sensor also enable efficient adsorption of the analyte molecule, further contributing to the charge transfer between the analyte (SARS-CoV-2) and the sensor [23]. The three-dimensional arrangement also reduces background fluorescence from the biological samples, enabling direct detection from saliva samples [25]. 

The SERS fingerprint obtained is then classified through a pre-trained machine learning algorithm, which provides a highly accurate diagnosis. The label-free SARS-CoV-2 saliva test utilizes a direct detection method from the saliva, thereby eliminating the need for complex isolation processes. Additionally, direct detection has multiple advantages, including substantially reducing the assay time, limiting the degradation of the sample, and providing an amplification-free complete SERS fingerprint for diagnosis. Notably, this method has approximately 10 min of sample-to-answer time, employs minimal reagents, can be easily automated for high-throughput testing, and uses a portable Raman spectrometer to amplify the potential use in a clinical setting. 

The nanostructured COVID sensor was synthesized using a physical synthesis process of multiphoton ionization. When an ultrashort pulsed laser strikes a solid single crystalline silicon wafer, it results in the formation of a nanostructured COVID sensor with unique chemical composition through intricate physical and chemical rearrangement processes aided by the drastic increase in temperature, leading to the vaporization of the Si ions and the creation of a high-temperature plasma plume. The plasma plume primarily comprises Si ions (evaporated from the Si), Oxygen ions from the atmospheric air. The pileup of ionic species in the plume leads to nanoparticle cluster formation condensing on the Surface of the silicon substrate. The representative TEM image in Figure 2b shows the cluster of nanostructured COVID sensor and their irregular morphology, with a mean particle size distribution of 5.2 nm (Figure 2c). 

The interaction of femtosecond laser pulses with Si substrate boosts the photon absorption efficiency, enabling their application for optical diagnostic techniques such as SERS and fluorescent imaging. The optical properties of the nanostructured COVID sensor were characterized using the fluorescence absorption properties shown in Figure 2d. The fluorescent spectra of the nanostructured COVID sensor are shown in Figure 2d. It can be observed from the spectra that nanostructured COVID sensors exhibit an absorption peak at 547 nm. The spectral characteristics result from the dependencies of photon emission on size, bandgap, optical density, and absorption cross–section, thus confirming the excitonic origin of the photoluminescence. The unique optical characteristics of the nanostructured COVID sensor are especially intriguing because it results in efficient charge-transfer reactions, which makes the nanostructured COVID sensor suitable for SERS-based diagnosis. 

To investigate the ability of the nanostructured sensor to differentiate between SARS-CoV-2 positive and SARS-CoV-2 negative saliva, we analyzed the SERS fingerprint of the individual components of the SARS-CoV-2. Structurally, the SARS-CoV-2 virus comprises spike protein, nucleocapsid protein, and single-stranded RNA as genetic material [26] and is approximately 100 nm in size; hence is detectable by SERS. Therefore, viruses can generate detectable SERS signals when the surface molecules contact the nanostructured sensor [27]. Figure 3a shows the representative SERS spectra of SARS-CoV-2 RNA. In addition, it can be observed that the spectra show the typical SERS peaks associated with single-stranded RNA, consistent with the reported literature [9]. 

The analyte molecule, such as RNA and proteins, are dropped on the Surface of the nanostructured sensor. When the Raman laser interacts with the nanostructured sensor and the analyte molecule, the analyte molecule produces characteristic vibrations, resulting in the analyte molecule’s unique spectral signature. For example, in the case of spike protein, the unique spectral signature is predominantly contributed by the tryptophan residues and the protein structural vibrations associated with the ACE2 receptor [28]. Similarly, the SERS spectral signature of nucleocapsid protein is contributed by the molecular vibrations of phenylalanine residues and the protein structural vibrations. Based on the observations mentioned above, it can be concluded that the adsorption of analyte molecules on the Surface of the nanostructured sensor produces a charge transfer reaction, which enhances the signals of the molecular vibrations resulting in the unique spectral signature of components of SARS-CoV-2 virus. 

Next, to ascertain the sensitivity of the nanostructured sensor, we studied the limit of detection for RNA, spike protein, and nucleocapsid protein. It can be observed from Appendix A that there is a linear relationship between the concentration of the analyte (RNA, spike protein, nucleocapsid protein) and the observed SERS intensity. For RNA, we observed a strong linear positive correlation between the concentration of RNA and the SERS peak intensity (Appendix A) with R^2^ provided in Appendix A. 

After establishing the positive correlation between the concentration of viral proteins, RNA, and the SERS peak intensity, it can be concluded that the nanostructured sensor possesses the ultra-sensitivity required for a rapid test for SARS-CoV-2. Furthermore, we optimized the sensor by utilizing the repeatability and reproducibility of the test. When considering a SERS-based assay for clinical applications, homogeneity of the SERS signal is one of the primary requirements [29]. As shown in Appendix A, the relative standard deviation of the characteristic peaks of RNA, spike protein, and nucleocapsid protein is <10%, showcasing a highly reproducible and uniform signal. In addition to the reproducibility, another parameter to consider for improving the assay’s sensitivity is minimizing the signal–to–noise ratio [30], as shown in Appendix A. These results suggest that the nanostructured sensor is better for detecting SARS-CoV-2 viruses with high sensitivity and reproducibility. 

Directly detecting SARS-CoV-2 infection from the saliva samples requires the ability to detect whole virus particles and the degraded viral components. Figure 3b shows the unique SERS signature of the whole virus particle, with peaks corresponding to the spike protein, RNA, and nucleocapsid protein at 721 cm^−1^ (peak 1), 1032 cm^−1^ (peak 2), 1270 cm^−1^ (peak 3), 1587 cm^−1^ (peak 4), 1643 cm^−1^ (peak 5) corresponding to Ribose, phenylalanine, Amide III, C-N stretching, Adenine (Ring mode) respectively [10,11,17,19]. 

We investigated the correlation between the variation in the number of viral particles/mL of saliva and the SERS intensity of the characteristic peaks. It can be observed from Figure 3c that there exists a strong linear correlation between the distinct peaks of SARS-CoV-2 (R^2^ = 0.7057). The correlation further validates the stability and reproducibility of this method. Moreover, it can be observed that the SERS peak intensity gradually decreases with decreasing viral titer. The linear calibration curve was fitted as Y = 0.0002250 × X + 3.490. Therefore, the limit of detection of SARS-CoV-2 viral particles in saliva was 50 viral particles per mL of saliva. 

Additionally, in the investigation of the similarities to assess the contribution of structural components of the whole viral particles to the SERS signature, we performed PCA analysis and hierarchical clustering analysis. It can be observed from Appendix A and Figure 3c that the SERS signature of the viral components and the whole virus show high similarity. The principal components show 94.3% similarity between whole virus, RNA, spike protein, and nucleocapsid protein. 

The analytical sensitivity of SARS-CoV-2 detection was determined by the limit of detection analysis, as reported in Figure 3. These results demonstrated a strong linear correlation between SERS intensity and viral load, which means that the sensitivity is viral load dependent. Studies have shown that molecular assays such as qRT-PCR and LAMP-PCR have an analytical sensitivity of ~100 copies of viral RNA/mL of transport media [31]. However, the viral load is directly related to the infectiousness and transmissibility of the virus [32,33]. Hence, it is essential to increase the analytical sensitivity of the assay to detect the presence of the SARS-CoV-2 virus at the early stages to reduce the transmission of the virus. 

Figure 3 shows the linear dependence of SERS intensity and the number of viral particles in saliva, demonstrating a highly sensitive detection. The high sensitivity of the nanostructured sensor indicates the involvement of multiple signal enhancement mechanisms. We performed fluorescence spectroscopy analysis to determine whether charge transfer between the nanostructured sensor’s surface and the virus’s surface is the primary mechanism of signal enhancement. It can be observed from Figure 4 that the characteristic absorption peak of the nanostructured sensor was at 547 nm. In contrast, the distinct peak of the whole virus particle is at 535 nm(Figure 4). Fluorescent mapping analysis revealed rapid fluorescence quenching when the viral particles were adsorbed onto the surface of the nanostructured sensor (Figure 4). However, the quenching is reduced with the number of adsorbed viral particles. It can be concluded from the concentration-dependent fluorescent quenching effect that charge transfer is the primary mechanism of enhancement. Studies have shown that the intrinsic fluorescent quenching in SERS systems results from surface plasmon-assisted charge transfer [34,35]. The charge transfer between the nanostructured sensor system and the viral particles is further enhanced by the ultrasmall size of the particles on the sensor [22]; the three-dimensional arrangement of the nanostructured sensor provides a high surface area for the adsorption of the viral particles [36,37,38]. In addition, the intrinsic functional groups on the surface of the nanostructured sensor provide efficient bonding for the adsorption of virus particles [39]. 

SARS-CoV-2 is a β coronavirus, which is symptomatically and structurally similar to other strains of viruses, such as 229E and OC43, that causes a tiny proportion of common cold and flu [1]; hence, cross-reactivity in diagnostic tests has been detected. A diagnostic test’s accuracy depends on reducing false positives by distinguishing structurally and symptomatically close viral strains. Here, we have analyzed the SERS signature of HCoV OC43, Influenza, and Respiratory Syncytial virus to investigate the assay’s specificity. 

The assay’s specificity was validated using three different viral strains, which present clinical symptoms such as SARS-CoV-2 viral infection and are structurally such as SARS-CoV-2. The specificity was analyzed using low pathological human coronaviruses, Influenza, and RSV viruses based on the seasonal prevalence [37]. Although the SERS assay showed high specificity for SARS-CoV-2 detection, additional studies to test the cross-reactivity with other common viral strains are required before large-scale deployment to high-risk areas and a wide range of population screening.

Figure 5 shows the representative characteristic SERS spectra of different viral strains. Through comparison of SERS spectra, the differences in the distinct peaks are 1165 cm^−1^, 1260 cm^−1^, and 1480 cm^−1^ corresponding to tyrosine, CH_2_ in-plane deformation, Amide II for Influenza [40,41], 1066 cm^−1^, 835 cm^−1^ corresponding to C-N stretching vibration, tyrosine for RSV [42], 663 cm^−1^, 902 cm^−1^, 1125 cm^−1^, 1333 cm^−1^ corresponding to valine/glutamine, Proline, C- N stretching, Amide III (β sheet) for human coronavirus OC43 [43]. We also performed a Partial least square analysis (Figure 6b) to determine the significant variables for prediction ability as represented by the regression coefficients. It can be observed from Figure 5b that the regression scores for different viral strains are significantly different, as observed through one- way ANOVA test (*p* = 0.0003). Additionally, we performed PCA analysis to validate the differences between the different viral strains. The heatmap in Figure 6d shows that PC1 shows a maximum variance of (97.8%) between the different viral strains. Owing to the close structural similarity between the viral strains, we utilized the tSNE algorithm to visualize the PC scores (Figure 5c), clearly distinguishing between different viral strains [11]. These results suggest that in addition to detecting the SARS-CoV-2 virus from a saliva sample, this approach can also identify the viral strain causing the infection. The method’s reliability is further validated by the consistency of the results using partial least square regression analysis and PCA analysis. 

We validated our protocol with clinical samples obtained from individuals who tested positive for SARS-CoV-2 through an RT-PCR test, negative saliva samples from asymptomatic individuals, and a simple machine learning algorithm. The machine-learning algorithm was trained using a negative saliva sample and simulated a positive saliva sample by spiking the saliva with a controlled number of viral particles. Figure 6a shows the diagnosis of SARS-CoV-2 directly from a saliva sample. It can be observed that the machine learning algorithm shows a highly accurate classification between positive and negative saliva samples. The classification accuracy was observed to be 95.6%.

Further, the sensitivity of the classification was found to be 90.5%, and 100% specificity was observed. The analysis of the effect of principal components on the model output showed that the PC1 contributed to the highly accurate classification. The predominant peaks of PC1 were 721 cm^−1^, 1032 cm^−1^, and 1270 cm^−1^. It suggests that the peaks correspond to viral RNA and spike protein. 

We also validated our protocol to ascertain the ability to distinguish between different viral strains in saliva samples. We utilized a decision tree algorithm to diagnose the presence of different viral strains. It can be observed from Figure 6b that the decision tree algorithm was able to distinguish different viral strains directly from saliva with high accuracy of ~98%. However, Figure 6b shows that one influenza sample was misclassified as RSV, which increased the false positive rate to 5.3% for RSV. This misclassification could further be improved by increasing the sample size, which could improve the algorithm. The decision tree in Figure 6b shows that PC1 contributes to accurately classifying different viral strains. 

We also validated the SERS-based assay with conventional RT- PCR assay. The results are presented in Figure 6c. First, using the SERS assay, a linear regression algorithm was applied to predict the viral load. It can be observed that the predicted vs. actual scatter plot follows a strong linear relationship. Next, we compared log 10 (viral load) as calculated by RT PCR assay with the viral load calculated using SERS assay. It can be observed that the values from the SERS assay are within the standard error margin of the viral load calculated using the RT-PCR assay. Furthermore, we performed a student *t*-test and revealed that the assay’s quantification values do not show any statistically significant differences (*p* = 0.1855), thereby confirming the validity of the SERS assay to be applied to clinical testing. 

Despite the several advantages of the SERS assay described in the study, some challenges still need to be addressed before extending the technique from the bench to a large-scale population setting. For example, in real-time, several samples could be used to test for the presence of the virus, and the viral load could differ in different sample sources. Hence, a standardized testing protocol should be developed for the assay. Further, the SERS system should distinguish between various viruses and other microbial species such as SARS-CoV-2, MERS, Influenza A, Adenovirus, and parainfluenza virus before deploying for clinical applications [44,45]. This study demonstrates the ability to distinguish between four closely related viral species; however, the number is much higher. Additionally, the SERS sensor should be able to identify positive and negative cases to meet regulatory requirements.

## 4. Conclusions

Viral RNA is primarily detected using nucleic acid- hybridization techniques such as RT-PCR and RT-LAMP. Detecting viral nucleic acid using RT-PCR was one of the most quickly established laboratory diagnosis methods in a novel viral pandemic. The technique primarily relies on amplifying small amounts of viral RNA in the patient sample. Besides, using samples with a low virus titer often amplifies the human genetic material rather than the viral genetic material, which often leads to a false-negative result. In addition, these processing steps also involve incorporating different benchtop equipment, reagents, and technical expertise during operation. Apart from the disadvantages mentioned above, the test results can take days to be generated. Besides, the clinical observations in various countries have revealed that the sensitivity of the RT-PCR detection approach is only 83.3%. Another significant drawback of RT-PCR is that the technique cannot be applied to test a large population. Several test kits could process a maximum of 2000 samples/per day. The method described here provides a rapid testing platform without extracting viral components. This “extraction–free” method eliminates the need for complex isolation procedures, which is prone to error. Further, it is not possible to detect low viral loads in PCR; hence, it cannot be used for early detection/diagnosis of the viral infection. On the other hand, the SERS-based label-free test described here has a limit of detection of 50 viral copies/mL of saliva, making it a viable test for large-scale screening of asymptomatic and presymptomatic individuals. Moreover, the label-free approach reduces the false- positives and enables adaptation to any novel pathogens or distinguishes new mutations in the existing pathogens, thereby providing a universal pathogen detection platform. Additionally, further analysis is required to optimize the sensor on other fronts for accurate quantification of viral load in the test samples. The sensitivity of the sensor could be improved by adding a sputtered layer of gold so that it can further enhance the signals from the viral components, and this methodology could be extended for use with many types of samples to improve the quantitative ability of the sensor. 

## Figures and Tables

**Figure 1 bioengineering-10-00391-f001:**
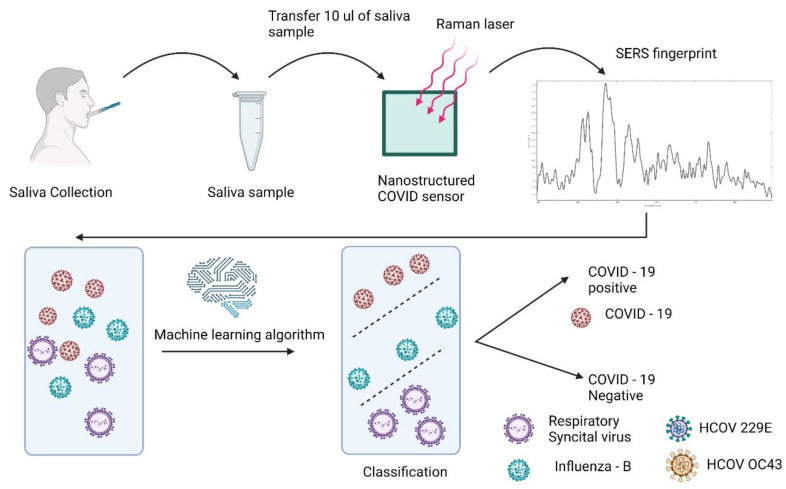
Schematic representation of the label-free rapid SARS-CoV 2 saliva test.

**Figure 2 bioengineering-10-00391-f002:**
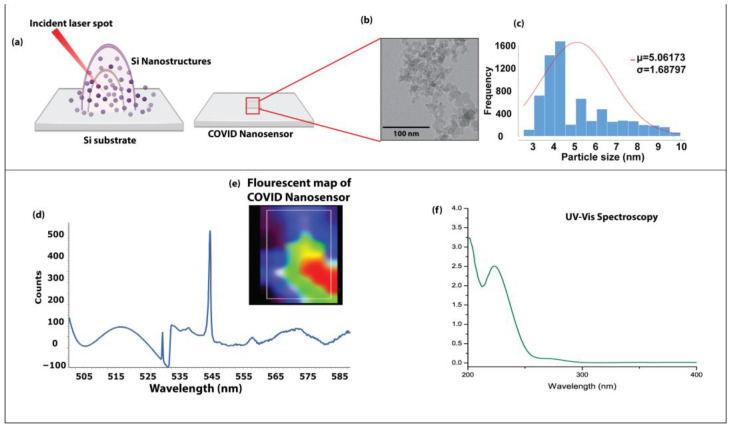
Synthesis and characterization of nanostructured COVID sensor (**a**) Schematic representation of the synthesis of nanostructured COVID sensor using multiphoton ionization process (**b**) Representative TEM image showing the morphology of quantum immunoprobe (Scale 50 nm) (**c**) Histogram showcasing the ultrasmall size of nanostructured COVID sensor based on TEM imaging, (**d**) Optical characterization of nanostructured COVID sensor using fluorescent spectroscopy and UV visible spectroscopy (**e**) Fluorescence mapping of the COVID nanosensor (**f**) UV-Vis spectra of the nanosensor.

**Figure 3 bioengineering-10-00391-f003:**
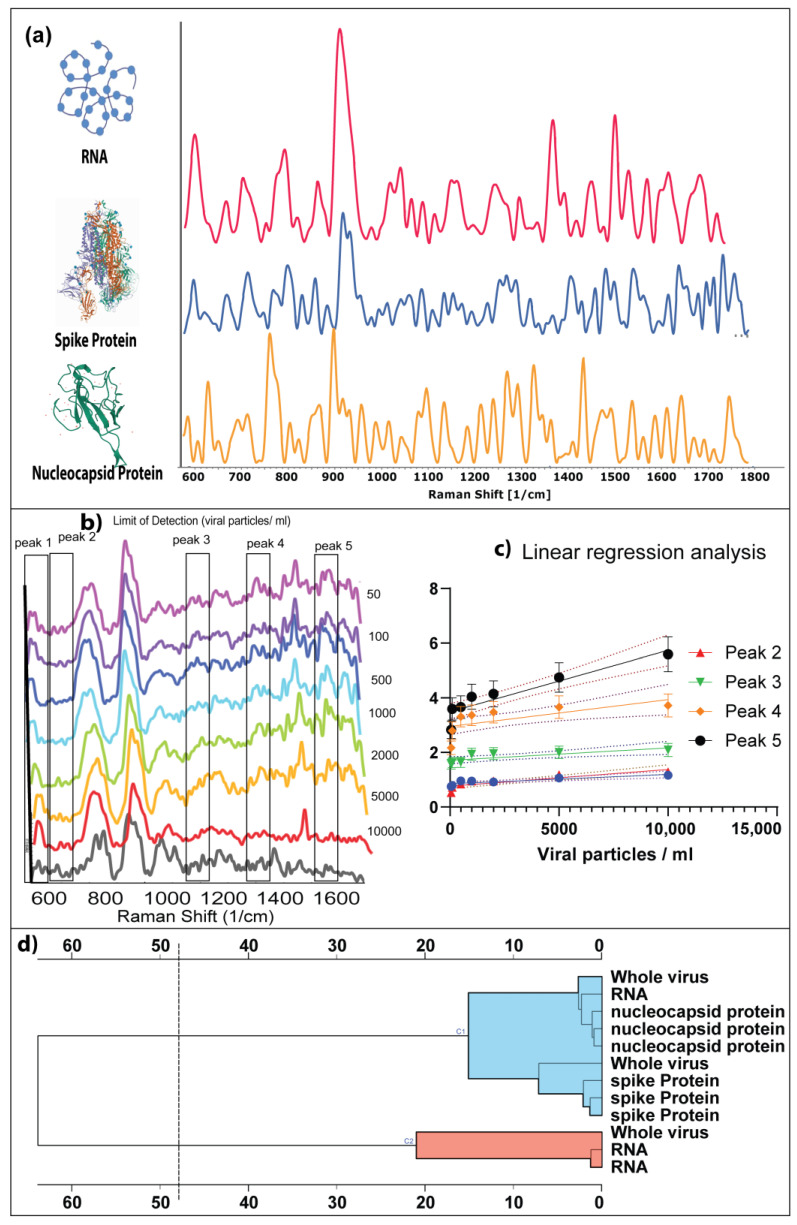
(**a**) Establishing the unique SERS signature of SARS-CoV-2 viral components. Analytical sensitivity of Label-free SERS assay by determining the limit of detection of the nanostructured sensor (**b**) SERS spectra of whole virus particles spiked in saliva varying from 10,000 particles to 50 particles per ml of saliva (**c**) Linear regression analysis based on the limit of detection (**d**) Hierarchical clustering analysis showing the cluster similarities between whole virus particles and their structural components.

**Figure 4 bioengineering-10-00391-f004:**
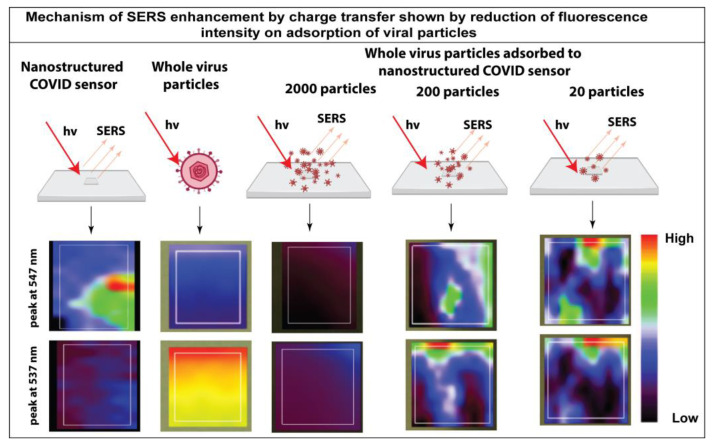
Mechanism of signal enhancement in the label-free SERS assay demonstrated through the fluorescence quenching on adsorption of whole virus particles on the nanostructured sensor.

**Figure 5 bioengineering-10-00391-f005:**
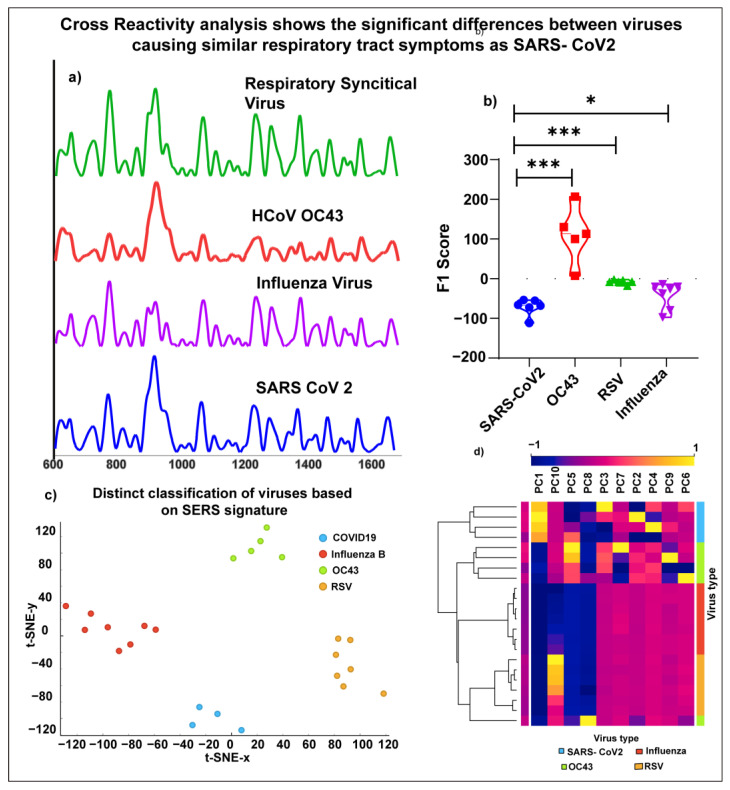
Cross-reactivity analysis shows the ability of the nanostructured sensor to distinguish different viruses clearly (**a**) Representative SERS spectra of SARS-CoV-2, Influenza, HCoV OC43, Respiratory syncytial virus, (**b**) Violin plot showing the F1 scores using partial least square method (**c**) tSNE analysis showing the clear distinction between different viral species, showing the high specificity of the label-free SERS saliva test using a nanostructured sensor (**d**) Heatmap showing the significant principal component contributing to the high specificity of the label-free SERS saliva test using the nanostructured sensor.(*—(*p*-value < 0.01), ***—(*p*-value < 0.0001)).

**Figure 6 bioengineering-10-00391-f006:**
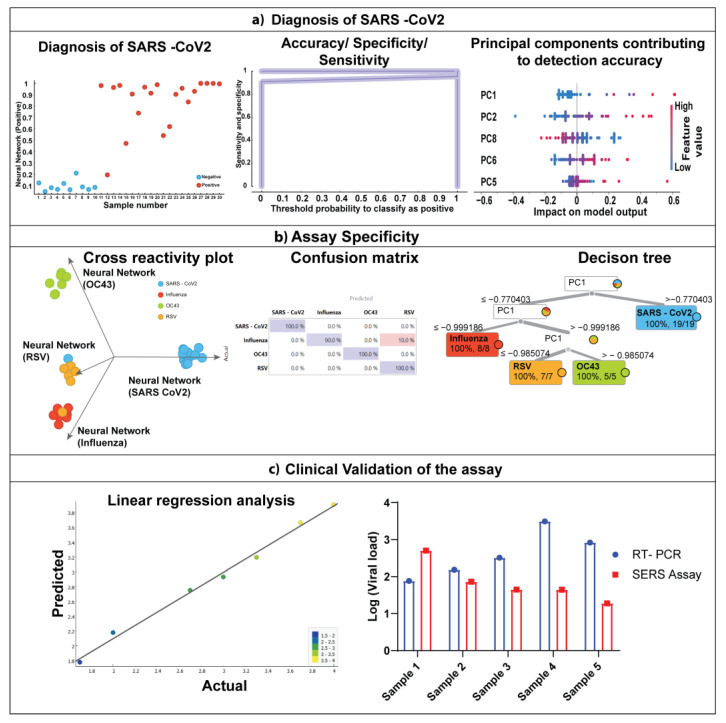
Direct diagnosis of SARS-CoV-2 virus from saliva samples (**a**) diagnostic parameters of COVID detection using Nanosensors (**b**) Applicability of nanostructured sensor for differentiating similar viruses (**c**) Clinical validation of the assay with RT-PCR.

## Data Availability

All data supporting the findings in this study are available from the corresponding author upon reasonable request.

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
