# Peer review of "Label-Free Saliva Test for Rapid Detection of Coronavirus Using Nanosensor-Enabled SERS"

_bioengineering, 2023, doi:10.3390/bioengineering10030391_

Round 1

Reviewer 1 Report

In this manuscript, the authors presented a SERS-based label free approach for detecting and distinguishing different respitory viruses in saliva samples. Overall, the approach is very attractive and I believe this manuscript can be published after minor revision.

Here are some problems that the authors need the address carefully in the revision.

1. The authors claim that the specificity is 100%. However, the peaks in the spectrums are very similar, at least in my eyes. Please carefully introduce how did you identify the characteristic peaks from such messy spectrums. 

2. Have you ever test real(pure) viral particles with your systems? Are the signals directly coming from the viral particles (or the immune response)? Please elaborate it in your manuscript. 

3. Have you tried any type of sample other than saliva?  

BTW, there may be an error in the author list (the name of the corresponding author is not in the list.

Author Response

Reviewer 1

In this manuscript, the authors presented a SERS-based label free approach for detecting and distinguishing different respiratory viruses in saliva samples. Overall, the approach is very attractive and this manuscript can be published after minor revision.

Response: The authors thank the reviewer for the positive comments on the manuscript. The authors have made the following changes in the manuscript to address the concerns and questions of the reviewer:

  • Additional explanation on identifying characteristic SERS peaks has been provided in the materials and methods section of the manuscript.
  • Elaborate explanation of the type of viral particles used for investigating the characteristic SERS peak of SARS -CoV2 has been incorporated in the manuscript.

Here are some problems that the authors need the address carefully in the revision.

Comment 1: The authors claim that the specificity is 100%. However, the peaks in the spectrums are very similar, at least in my eyes. Please carefully introduce how did you identify the characteristic peaks from such messy spectrums.

Response: The authors thank the reviewer for the valuable comment. The characteristic SERS peaks from the viral RNA, protein, and nucleocapsid peaks used for analysis were carefully chosen by the in-built B&W Tek software, with the signal- to-noise ratio higher than three times the standard deviation of the data. The reproducibility of the signals was maintained by keeping the acquisition parameters constant for all Raman measurements. criteria for peak selection have been incorporated in the materials and methods section of the manuscript as shown below:

“The SERS peaks used for analysis were carefully chosen with the signal- to-noise ratio higher than three times the standard deviation of the data. Further, to ensure the reproducibility and uniformity of signals, the acquisition time was kept constant at 10s, and three acquisitions were made.”

Comment 2: Have you ever tested real(pure) viral particles with your systems? Are the signals directly coming from the viral particles (or the immune response)? Please elaborate it in your manuscript.

Response: The authors thank the reviewer for the comment. Purified viral particles obtained from ATCC were used for obtaining the SERS spectra of the SARS-CoV-2 virus, as shown in figure 3. Further, the signals obtained were coming directly from the viral particles and not from the immune response, as no cells were involved in the process. The following explanation in the manuscript elucidates the SERS acquisition from the viral particles:

“To investigate the ability of the nanostructured sensor to differentiate between SARS – CoV-2 positive and SARS – CoV 2 negative saliva, we analyzed the SERS fingerprint of the individual components of the SARS- CoV2 virus. Structurally, the SARS- CoV 2 virus comprises spike protein, nucleocapsid protein, and single-stranded RNA as genetic material [27] and is approximately 100nm in size; hence is detectable by SERS. Therefore, viruses can generate detectable SERS signals when the surface molecules contact the nanostructured sensor [28]. Figure 3a shows the representative SERS spectra of SARS- CoV 2 RNA. In addition, it can be observed that the spectra show the typical SERS peaks associated with single-stranded RNA, consistent with the reported literature [9].”

Comment 3: Have you tried any type of sample other than saliva? 

Response: The authors appreciate the reviewer for the question. However, we have performed the test only with saliva in this study to demonstrate the applicability of the test for a larger population screening. Further, the test methodology is so versatile that it can be adapted to test multiple analytes, including nasal swabs, blood, plasma, and other bodily fluids. The selection of sample type was made considering factors like the ease of sample collection, availability, storage, and handling, making it minimally or non-invasive. Further, the study aims at a larger population screening, and therefore, saliva samples were considered the most suitable sample type for the test.

Reviewer 2 Report

This study presents a method for detecting SARS-CoV-2 using SERS. It has the advantage of being able to detect SARS-CoV-2 using a small amount of sample in a short time, but it has the disadvantage of requiring specialized equipment for SERS. This disadvantage can be compensated by machine learning algorithms, but it will be difficult to develop a biosensor that can be used for point-of-care testing. The following advice will help in the qualitative development of this paper, and I hope that this thesis will help in the development of point-of-care testing biosensors using SERS.

1. Add a cover page in the Supplementary Materials with the title of the paper, author names, etc.

2. Use different types of DASHs in their proper roles.

3. The notation of SRAS-CoV-2 is not uniform. Use only one notation.

4. Explain in detail the 'Machine learning algorithm' in Figure 1.

5. Explain the difference between the results obtained by RT-PCR and SERS assay for samples 1-6 in Figure 6c.

Author Response

Reviewer 2

This study presents a method for detecting SARS-CoV-2 using SERS. It has the advantage of being able to detect SARS-CoV-2 using a small amount of sample in a short time, but it has the disadvantage of requiring specialized equipment for SERS. This disadvantage can be compensated by machine learning algorithms, but it will be difficult to develop a biosensor that can be used for point-of-care testing. The following advice will help in the qualitative development of this paper, and I hope that this thesis will help in the development of point-of-care testing biosensors using SERS.

Response: The authors thank the reviewer for the positive comment on the novelty of the manuscript. The authors have made the following changes in the manuscript to address the questions and concerns of the reviewer:

  • Additional explanation on the machine learning algorithm has been incorporated in the manuscript to provide improve the clarity of the manuscript.
  • The conclusion section of the manuscript has been modified to address the need for an extensive study to assess the quantitative ability of the sensor

The point–by–point response is as follows:

Comment 1. Add a cover page in the Supplementary Materials with the paper’s title, author names, etc.

Response: The authors have incorporated the necessary changes in the supplementary section of the manuscript.

Comment 2. Use different types of DASHs in their proper roles.

Response: The authors thank the reviewer for carefully examining the manuscript. Necessary changes have been made the in the manuscript as per the reviewer’s advice.

Comment 3. The notation of SARS-CoV-2 is not uniform. Use only one notation.

Response: The authors thank the reviewer for noticing the notation. The authors have changed the correct notation of SARS-CoV-2 throughout the manuscript per the reviewer’s advice.

Comment 4. Explain in detail the 'Machine learning algorithm' in Figure 1.

Response: The authors thank the reviewer for the comment. The authors have explained the machine learning algorithm in detail to improve the reproducibility and clarity of the study. The authors have incorporated the following in the materials and methods section of the manuscript:

“Machine Learning Algorithm:

The preprocessed SERS spectra were dimensionally reduced to principal components using PCA. The PCs with maximum variances were used to train the decision tree algorithm. The decision tree algorithm splits the data into nodes based on class purity. The decision tree parameters used for the classification were: the minimum number of instances in the leaves was maintained at 3, and the maximal tree depth was maintained at 5 to limit non-specific sub-set classification. Further, the algorithm was instructed to stop classification after the majority threshold of 95% was reached. The machine learning algorithm’s performance was assessed using the calibrated probabilities of classification. The classification threshold was maintained at 0.5.”

Comment 5. Explain the difference between the results obtained by RT-PCR and SERS assay for samples 1-6 in Figure 6c.

Response: The authors thank the reviewers for the thoughtful comment. Statistical analysis has shown that the difference is insignificant between the RT-PCR and SERS assay results shown in figure 6c. Further study will be extended to evaluate and optimize the quantitative capability of the sensor to detect the quantity of the viral load in the sample. The following has been incorporated in the conclusion section of the manuscript to improve its clarity of the manuscript:

“Additionally, further analysis is required to optimize the accurate quantification of viral load in the test samples using the SERS sensor, which would be extended in our future studies to improve the quantitative ability of the sensor.”

Reviewer 3 Report

The authors have proposed a novel method to detect COVID-19. They presented the manuscript well and it can be published after applying comment blow. Therefore, in this statute, I ask for a minor revision. 

1: The authors should have compared the response linear range, time of response, and limit of detection with the previously reported biosensors and sensors related to COVID-19

2. That would be great if the roughness of the surface of the porous silicon have been studied with AFM. I think the sensitivity of the sensor would increase if a thin layer of gold was deposited on this surface of porous silicon by gold sputtering because of its good plasma resonance property. That is just advice.

3. I do not agree with the authors that the proposed technique is a POT sensor. Because of the size of the Raman device, the cost of the final sensor as a sensor, and the defiantly of the pretreatment process on the real sample, it can not be done by people and their homes like a pregnancy test kit or blood sugar test sensor.

4. It is better to explain the response mechanism of the sensor. what the plasma resonance is and how the target changes the property of the substrate.

Author Response

Reviewer 3

The authors have proposed a novel method to detect COVID-19. They presented the manuscript well, which can be published after applying comment blow. Therefore, in this statute, I ask for a minor revision.

Response: The authors thank the reviewer for the positive comment. The authors have included the following in the manuscript to address the reviewer’s questions and concerns:

  • New supplementary table S2 has been added to the manuscript to compare the performance of the nanostructured COVID sensor designed in this study with the previously reported COVID – 19 sensors
  • Additional references have been incorporated in the manuscript to improve the clarity of the manuscript

Comment 1: The authors should have compared the linear response range, time of response, and limit of detection with the previously reported biosensors and sensors related to COVID-19

Response: The authors thank the reviewers for their valuable insight. In addition, the authors have compared the technique with other previously reported sensors and methods related to COVID-19 and incorporated the following table S2 in supplementary information :

Table S2: List of Techniques/ Sensors showing their detection parameters compared to our technique.

Technique

Limit of detection

Time

Reference

Multiple antibody assay

1.1 × 105 copies/ml

NA

[1]

RT-PCR

3.8 & 5.2 copies per reaction

NA

[2]

Solsten SARS-CoV-2 Antigen Enzyme-Linked ImmunoSorbent Assay Kit

4.1 PFU/mL

NA

[3]

Elecsys SARS‐CoV‐2 Antigen ElectroChemiLuminescence ImmunoAssay

37 PFU/mL

NA

[3]

Simoa SARS CoV‐2 N Protein Advantage Kit

0.15 PFU/mL

NA

[3]

Recombinase polymerase amplification (RPA)

2 copies per sample

50 mins

[4]

Recombinase polymerase amplification (RPA)

2.5 copies/μl input

50 mins

[5]

LAMP (Fluoresence)

30 copies/μL (150 copies)

40 min

[6]

PCR (Lateral flow Assay)

13.5 copies/μL

65 mins

[7]

PER (Fluoresence)

1.3 pM

40 min

[8]

Nanostructured COVID Sensor

10 copies/ml

10 min

This study

The following references have been added to the manuscript:

  • Traugott, M.T.; Hoepler, W.; Seitz, T.; Baumgartner, S.; Karolyi, M.; Pawelka, E.; Friese, E.; Neuhold, S.; Kelani, H.; Thalhammer, F.; et al. Diagnosis of COVID-19 Using Multiple Antibody Assays in Two Cases with Negative PCR Results from Nasopharyngeal Swabs. Infection 2021, 49, 171–175, doi:10.1007/s15010-020-01497-2.
  • Corman, V.M.; Landt, O.; Kaiser, M.; Molenkamp, R.; Meijer, A.; Chu, D.K.; Bleicker, T.; Brünink, S.; Schneider, J.; Schmidt, M.L.; et al. Detection of 2019 Novel Coronavirus (2019-NCoV) by Real-Time RT-PCR. Eurosurveillance 2020, 25, 2000045, doi:10.2807/1560-7917.ES.2020.25.3.2000045.
  • Hillig, T.; Kristensen, J.R.; Brasen, C.L.; Brandslund, I.; Olsen, D.A.; Davidsen, C.; Madsen, J.S.; Jensen, C.A.; Hansen, Y.B.L.; Friis-Hansen, L. Sensitivity and Performance of Three Novel Quantitative Assays of SARS-CoV-2 Nucleoprotein in Blood. Sci Rep 2023, 13, 2868, doi:10.1038/s41598-023-29973-3.
  • Huang, Z.; Tian, D.; Liu, Y.; Lin, Z.; Lyon, C.J.; Lai, W.; Fusco, D.; Drouin, A.; Yin, X.; Hu, T.; et al. Ultra-Sensitive and High-Throughput CRISPR-p Owered COVID-19 Diagnosis. Biosens Bioelectron 2020, 164, 112316, doi:10.1016/j.bios.2020.112316.
  • Sun, Y.; Zhang, Q. Finite Element Analysis of Rock Breaking Experimental Bench of High Pressure Water Fracturing Assisted Pick. In Proceedings of the International Conference on Intelligent Equipment and Special Robots (ICIESR 2021); SPIE, December 12 2021; Vol. 12127, pp. 402–408.
  • Isothermal Amplification and Ambient Visualization in a Single Tube for the Detection of SARS-CoV-2 Using Loop-Mediated Amplification and CRISPR Technology | Analytical Chemistry Available online: https://pubs-acs-org.ezproxy.lib.torontomu.ca/doi/10.1021/acs.analchem.0c04047 (accessed on 5 March 2023).
  • Detection of SARS-CoV-2 with Solid-State CRISPR-Cas12a-Assisted Nanopores | Nano Letters Available online: https://pubs.acs.org/doi/10.1021/acs.nanolett.1c02974 (accessed on 5 March 2023).
  • A Simple and Rapid Method to Assay SARS-CoV-2 RNA Based on a Primer Exchange Reaction. Chemical Communications 2022, 58, 4484–4487, doi:10.1039/d2cc00488g.

Comment 2. That would be great if the roughness of the surface of the porous silicon have been studied with AFM. I think the sensitivity of the sensor would increase if a thin layer of gold was deposited on this surface of porous silicon by gold sputtering because of its good plasma resonance property. That is just advice.

Response: The authors thank the reviewer for the valuable suggestion. The authors agree with the reviewer that incorporating plasmonic components like a layer of gold would increase the sensitivity of the sensors by enhancing the SERS intensity of the spectra. However, the same would also contribute to the change in inherent structures of the biomolecules like proteins, nucleic acids, and sugars, which is detrimental to the precise identification of characteristic peaks and therefore, poor specificity. In addition, the addition of gold layer to the silicon layer critically affects the uniformity, reproducibility of the SERS signals, critical for identification of the viral particles. In addition, due to the sub 10 nm size of the silicon particles in the sensor, plasmonic resonance is enhanced manifolds eliminating the requirement for gold sputtering. The SERS enhancement by the plasmonic resonance due to the sub-10 nm size of the nanoparticles of the sensor is shown in figure 3 of the manuscript.

The following references corroborates the theory:

  • Anik, M. I.; Mahmud, N.; Al Masud, A.; Hasan, M. Gold Nanoparticles (GNPs) in Biomedical and Clinical Applications: A Review. Nano Select 2022, 3 (4), 792–828. https://doi.org/10.1002/nano.202100255.
  • Grant, S. A.; Spradling, C. S.; Grant, D. N.; Fox, D. B.; Jimenez, L.; Grant, D. A.; Rone, R. J. Assessment of the Biocompatibility and Stability of a Gold Nanoparticle Collagen Bioscaffold. Journal of Biomedical Materials Research Part A 2014, 102 (2), 332–339. https://doi.org/10.1002/jbm.a.34698.

Comment 3. I do not agree with the authors that the proposed technique is a POT sensor. Because of the size of the Raman device, the cost of the final sensor as a sensor, and the defiantly of the pretreatment process on the real sample, it cannot be done by people and their homes like a pregnancy test kit or blood sugar test sensor.

Response:

The authors thank the reviewer for the valuable insight. The overall objective of the study is to develop a device that enables population screening possible with a high throughput capacity to process a significantly large number of samples  in a short span of time. The most appropriate target location for this detection device would be highly populated public settings like airports, Hospitals, Malls and other common spaces. A device like this will enable a point-of-care COVID-19 screening of a substantially high number of samples in a short span of time.

Comment 4. It is better to explain the response mechanism of the sensor. what the plasma resonance is and how the target changes the property of the substrate.

Response: The authors thank the reviewer for the valuable comment. The authors have described the response mechanism of the sensor in Figure 4, where the interaction of viral particles change the fluorescence property of the sensor and the viral particles as demonstrated by quenching of fluorescence. The quenching effect is primarily due to the charge transfer interaction between the sensor and the viral particles. In addition, the decrease in fluorescence because of the charge transfer interaction, further amplifies the SERS response, resulting in high sensitivity.

Round 2

Reviewer 2 Report

 The manuscript has a scholarly content suitable for Bioengineering. The manuscript is well organized and specific. In addition, the  is expected to have a good impact on the advancement of academic and practical biosensors.

Author Response

Reviewer 2:

The authors have addressed almost all the issues requested by the reviewers. However, there are still minor issues to correct. For example:

Response: The authors thank the reviewer for the positive comments about the manuscript. The authors have made the following changes in the manuscript to address the questions and concerns of the reviewer:

  • Changes have been incorporated to the manuscript to change the POC sensor to a rapid diagnostic sensor as per the reviewer’s advice.
  • The conclusion section of the manuscript has been modified to address the need for additional gold sputtering to assess the enhancement efficiency of the sensor.
  • Repeated phrases have been removed from the manuscript.

The point–by–point response is as follows:

Comment 1: I agree with Reviewer #3  that the system proposed by the authors is not a POC system, since it requires a Raman microscope. A POC testing system implies measurements near or next to the patient and, moreover, to be carried out by untrained personnel. The sensor would need a portable Raman device, for example, to be considered a POC sensor. Although it is a device that would allow the measurement of a large number of samples, this characteristic does not imply that it is a POC device. So, please consider that perhaps in the future it could be a POC sensor. The paper by Liu et al., 2015. Surface Plasmon Resonance Biosensor Based on Smart Phone Platforms is a POC device. This is an example of a system that can be used near the patients. Of course, it makes SPR measurements, but the authors made a portable system to mae the measurements.

Response:

The authors thank the reviewer for the insightful comment. The authors would like to clarify that the study was conducted using a portable, hand-held, Raman device that can be easily transported and used to record spectral information from patient samples at the point of care. However, to remove the ambiguity, the authors have changed the POC into “rapid diagnosis” throughout the manuscript according to the reviewer’s advice. This makes our sensor a rapid COVID diagnostic tool.  

Comment 2:

I also agree with Reviewer #3's advise that the sensitivity of the sensor would increase if a thin layer of gold was deposited on this surface of porous silicon by gold sputtering, because of its good plasma resonance property. The authors respond that this would also contribute to the change in inherent structures of the biomolecules like proteins, nucleic acids, and sugars. I do not agree with the authors that gold changes the structure of macromolecules such as proteins. Perhaps of other much smaller and more flexible molecules, but it is unlikely. Of course, if gold is added, there could be changes in the SERS spectra that the authors record with silicon, since certain parts of the molecules would fix to the gold, seeing those parts intensified in the spectra, while the rest of the molecule (away from the surface) would not be seen in them. But this could improve the signal from the biosensor, and not be a disadvantage as the authors suggest.

Response:

The authors thank the reviewer for the valuable insight. The authors agree with the reviewers regarding the enhancement of Raman signals by the nanosensor after being sputtered with gold and its biocompatibility in retaining the inherent molecular structures. Considering the ease of fabrication, the sensors were used directly after the laser ablation technique that provided us the reported sensitivity, specificity, and accuracy of the diagnostic technique. However, to improve the sensitivity and limit of detection of the device, further optimization of the sensor will be performed. The optimization will be performed by adding sputtered gold on the sensor, test with additional samples and perform stability analysis to improve various aspects the sensor. This will improve the sensor’s ability to be used for detection of  multiple stages and variants of COVID.

The same has been incorporated in the conclusion section of the manuscript to improve the clarity of the manuscript as follows:

“Additionally, further analysis is required to optimize the sensor in other fronts for accurate quantification of viral load in the test samples. The sensitivity of the sensor could be improved by adding a sputtered layer of gold, so that it can further enhance the signals from the viral components and this methodology could be extended for use with many types of sampleto improve the quantitative ability of the sensor”

Comment 3:

In the conclusions there are two almost repeated sentences in the same paragraph (Detecting viral nucleic acid using RT-PCR was one of the most quickly established laboratory diagnosis methods in a novel viral pandemic... and ... Detecting viral nucleic acid using RT-PCR was one of the most quickly established laboratory diagnosis methods in a novel viral pandemic.). Rewrite for clarity without using the same expressions.

Response:  The authors thank the reviewer for noticing the repeated use of phrases in a sentence. The authors have carefully resolved the issue in the manuscript as per the reviewer’s suggestion.
